🔓 | **Open Peer Review** | Bacteriology | Research Article

# The secretome of the environmental bacterium *Pseudomonas protegens* PBL3 has broad-spectrum antimicrobial activity against plant and human pathogenic bacteria

Biwesh Ojha,[1,2] Kiyara Grosz,[3] Clemencia M. Rojas[1,4]

**ABSTRACT**   Antimicrobials are widely used to manage bacterial diseases in humans, pets, and livestock. Unfortunately, their extensive use has contributed to the emergence of antimicrobial resistance (AMR), a threat to global health. In contrast to the widespread antimicrobial use in medicine and veterinary, the use of antimicrobials to control bacterial diseases in plants is limited, posing a threat to global food security. These two threats highlight the urgent need to discover novel antimicrobial compounds with new modes of action. Previously, we demonstrated that the environmental bacterium *Pseudomonas protegens* PBL3 inhibits the growth of the plant pathogenic bacterium *Burkholderia glumae*, indicating that this strain produces and secretes potent antimicrobial compounds. In this study, we showed that the *P. protegens* PBL3 secretome also shows a broad spectrum of activity against plant pathogenic *Burkholderia* sp., as well as the human pathogens *Acinetobacter baumannii*, *Escherichia coli* O157:H7, *Listeria innocua*, *Enterococcus faecium,* and *Staphylococcus aureus,* with levels of growth inhibition ranging from 25% to 95% depending on the specific pathogen. These results highlight the *P. protegens* PBL3 secretome as a promising source of natural antimicrobial compounds with potential applications in both agriculture and clinical settings.

**IMPORTANCE**   This work shows that the environmental bacterium called *Pseudomonas protegens* PBL3 produces and secretes molecules that have a broad spectrum of activity against pathogenic bacteria causing important diseases in plants and humans. The use of the collection of molecules that *P. protegens* PBL3 produces and secretes (secretome) instead of single molecules is likely to be more effective, as this collection of molecules is expected to be of diverse chemical structure and mode of action, which makes it less likely that pathogenic bacteria develop resistance to them. These findings highlight the potential of beneficial microbes as a sustainable source of broad and effective antimicrobials for both agriculture and medicine.

**KEYWORDS**   antimicrobials, plant pathogens, human pathogens, bacteria

Antimicrobials to control bacterial diseases have long been used to manage bacterial infections in humans, pets, livestock, and aquatic animals as therapeutic or prophylactic agents (1–4). However, this widespread use of antimicrobials has significantly contributed to the emergence of antimicrobial resistance (AMR) (4). The AMR threat is becoming a matter of global concern that has not been mitigated, as most of the recently approved antimicrobials are derivatives of previously known classes (5, 6), and bacteria quickly evolve mechanisms to render them ineffective. In contrast to the widespread use of antimicrobials in the medical and veterinary fields, the use of antimicrobials to combat bacterial diseases in crops has been limited, as effective antimicrobials for those pathogens have not been identified, have proven ineffective, or

**Peer Reviewer** Liezel del Castillo Atole, Partido State University, Goa, Camarines Sur, Philippines

Address correspondence to Clemencia M. Rojas, crojas2@unl.edu.

The authors declare no conflict of interest.

See the funding table on p. 8.

have only been approved for agricultural use for a limited number of bacterial pathogens (7, 8). Thus, both the threat of antimicrobial resistance in the medical and veterinary fields and the absence of effective antimicrobials in agriculture necessitate the discovery of new molecules with unique chemistries and modes of action.

Historically, naturally occurring microorganisms have been a prolific source of antimicrobial compounds since the discovery of penicillin from the fungus *Penicillium notatum* that marked the beginning of the antimicrobial era and demonstrated the therapeutic potential of microbial metabolites (9). Since then, various fungal species have yielded diverse classes of antimicrobials. Beyond fungi, bacteria, particularly actinomycetes, have played a dominant role in antimicrobial discovery. Of the estimated 22,500 biologically active microbial compounds, more than 64% of all known antimicrobial classes are derived from the genus *Streptomyces* (10, 11). Additional contributors of antimicrobials are bacteria from the genera *Bacillus*, *Paenibacillus*, and *Pseudomonas* (11). Although natural products from microorganisms have yielded the majority of clinically used antimicrobials, a vast reservoir of untapped bioactive compounds still exists in microbial genomes.

Previously, the environmental bacterium *Pseudomonas protegens* PBL3 was identified as an antagonistic bacterium to the plant pathogenic bacterium *Burkholderia glumae*, the causal agent of the rice disease Bacterial Panicle Blight (12, 13). The antagonistic activity of *P. protegens* PBL3 against *B. glumae* was associated with the *P. protegens* PBL3 secreted fraction (secretome), indicating that *P. protegens* PBL3 produces and secretes antimicrobials (13). Those antimicrobials are likely associated with at least 14 biosynthetic clusters in the *P. protegens* PBL3 genome predicted to encode secondary metabolites, such as 2 Pyoverdines, Pyrrolnitrin, Pyoluteorin, 2,4-diacetylphloroglucinol, Orfamide C, Arylpolyene, 2-Hydroxyphenylthiazoline/Thiazostatin/Watasemycin B/ Pyochelin, Fengycin, Lipopeptide 8D1-1, 2 Bacteriocins, Cyclodipeptides, and N-acetyl-glutamine amide (13). Among those, eight were commercially available for testing, but only Pyoverdine, Pyoluteorin, and 2,4-Diacetylphloroglucinol inhibited the growth of *B. glumae* at high concentrations, yet did not fully explain the antimicrobial activity of *P. protegens* PBL3 secretome (14). Moreover, other studies demonstrated the antifungal, but not antibacterial activity of Thiazostatin, Orfamide, and Fengycin (15–18); and only Watasemycin B has demonstrated weak antibacterial activity (17).

Although the complete chemical composition of the *P. protegens* PBL3 secretome is still unknown, the predicted complexity based on its coding capacity led us to hypothesize that the *P. protegens* PBL3 secretome possesses broad-spectrum antimicrobial activity against other bacterial pathogens. In this study, we investigated the spectrum of activity of the *P. protegens* PBL3 secretome against plant and human pathogenic bacteria. Our results showed that the secretome inhibited the growth of plant pathogenic bacteria of the genera *Burkholderia,* as well as selected human pathogens, such as *Acinetobacter baumannii*, *Yersinia enterocolitica*, *Escherichia coli* O157:H7, *Listeria innocua*, *Enterococcus faecium, and Staphylococcus aureus*. These findings suggest that the *P. protegens* PBL3 secretome represents a source of antimicrobials of broad spectrum and possibly novel mechanisms of action for the management of both plant and human bacterial pathogens.

## MATERIALS AND METHODS

### Bacterial strains

Bacterial strains in this study are listed in Table 1. *P. protegens* PBL3 was grown on LB agar, plant pathogens were grown on King's B (KB) agar, and human pathogens were grown on Tryptic Soy agar. Bacterial strains were streaked from the −80°C freezer on agar plates and incubated at 28°C for plant pathogenic bacteria and 37°C for human pathogenic bacteria.

**TABLE 1** Bacterial strains used in this study

| Bacterial strains | Reference/source |
|---|---|
| *Acinetobacter baumannii* | Martin Conda-Sheridan, University of Nebraska Medical Center |
| *Burkholderia cenocepacia* CU0318 | (19) |
| *Burkholderia cenocepacia* CU3094 | (19) |
| *Burkholderia cenocepacia* CU3368 | (19) |
| *Burkholderia cenocepacia* CU3370 | (19) |
| *Burkholderia cenocepacia* CU3371-1 | (19) |
| *Burkholderia cenocepacia* CU3371-2 | (19) |
| *Burkholderia cenocepacia* CU6878 | (19) |
| *Burkholderia gladioli* CU3082 | (19) |
| *Burkholderia gladioli* CU3083 | (19) |
| *Burkholderia gladioli* CU3891 | (19) |
| *Burkholderia glumae* UAPB10 | Yeshi Wamishe, University of Arkansas, Fayetteville |
| *Burkholderia glumae* UAPB11 | Yeshi Wamishe, University of Arkansas, Fayetteville |
| *Burkholderia glumae* UAPB13 | Yeshi Wamishe, University of Arkansas, Fayetteville |
| *Burkholderia* sp. *O64a* | Steven Beer, Cornell University |
| *Enterococcus faecium* | Food Processing Center, University of Nebraska-Lincoln |
| *Erwinia amylovora* | Elena Garcia, University of Arkansas, Fayetteville |
| *Escherichia coli* K 12 | ATCC 10798 |
| *Escherichia coli* O157: H7 | U.S. Department of Agriculture-Agriculture Research Service |
| *Listeria innocua* | ATCC 33090 |
| *Pseudomonas aeruginosa* PAO1 | Jim Alfano, University of Nebraska-Lincoln |
| *Pseudomonas aeruginosa* PA103 | Jim Alfano, University of Nebraska-Lincoln |
| *Pseudomonas protegens* PBL3 | (13) |
| *Salmonella typhi* | Food Processing Center, University of Nebraska-Lincoln |
| *Staphylococcus aureus* | ATCC 25923 |
| *Xanthomonas axonopodis* | Craig Rothrock, University of Arkansas, Fayetteville |
| *Yersinia enterocolitica* | Jim Alfano, University of Nebraska-Lincoln |

## Antimicrobial assay against plant pathogenic bacteria

A single colony of *P. protegens* PBL3 was grown in M9 Minimal media (1× M9 Minimal Salts [Sigma-Aldrich, St. Louis, MO, U.S.A.] 2 mM $MgSO_4 \cdot 7H_2O$, 56 mM myo-inositol, and 0.2 mM $CaCl_2$) for 36–48 h until the optical density measured at 600 nm ($OD_{600}$) reached 0.5. The secreted fraction (secretome) was separated from whole bacterial cells by centrifugation and filter sterilized as previously described (14).

Plant pathogenic bacterial strains were scraped from the agar plates and resuspended in sterile water to a final concentration of $OD_{600} = 0.2$. The bacterial suspension was mixed with either *P. protegens* PBL3 secretome or water (control) and incubated on a shaker for 18 h. Bacterial growth was evaluated by measuring the $OD_{600}$. Each treatment had three replications and was repeated three times.

## Antimicrobial assay against human pathogenic bacteria

Human pathogenic bacterial strains were scraped from the agar plates and resuspended in 1× phosphate-buffered saline (PBS) to an $OD_{600}$ of 0.2 ($1.5 \times 10^8$ colony-forming units [CFU]/mL) and further diluted in a growing medium to a final bacterial concentration of $1.5 \times 10^6$ CFU/mL. The antimicrobial activity of the *P. protegens* PBL3 secretome was assessed in a 96-well microtiter plate. Each well contained a total volume of 100 µL, consisting of growth media, *P. protegens* PBL3 secretome (or PBS control), and the bacterial suspension. A fixed volume of 10 µL of the bacterial suspension to be evaluated was added to both treatment and control wells. The antimicrobial potency of *P. protegens* PBL3 was evaluated using a microdilution assay, where varying volumes (10, 20, and 30 µL) of the PBL3 secretome were tested, adjusting the final volume to 100 µL with growth media. Each treatment was conducted in six replicates. For the controls, PBS was

added in volumes equivalent to those of the secretome. Growth media alone (100 µL) was used as a negative control, while 10 µL of bacterial suspension in 90 µL of growth media was used as a positive control. Microtiter plates were incubated at 37°C for 18 h. Visual confirmation of bacterial growth was done by adding 0.1% of 2,3,5-triphenyl-tetrazolium chloride (TTC) solution to each well, followed by incubation at 37°C for 30 min to quantify the conversion of colorless TTC into formazan that occurs through the metabolic activities of live bacteria. The formation of formazan was measured by reading absorbance at 470 nm (A470nm) using a plate reader (Synergy 2 Multi-mode microplate reader, Biotek Instruments, Inc.).

## Statistical analysis

Data analysis was conducted using R Studio version 4.2.1, calculating the mean and standard deviation of bacterial growth measured for both *P. protegens* PBL3 treatment and control groups in plant and human pathogens. An unpaired two-way Student's *t*-test was performed to determine differences between the control and *P. protegens* PBL3 secretome, with the significance level set at $P = 0.05$.

## RESULTS

### The *P. protegens* PBL3 secretome has a broad-spectrum activity against plant pathogens

The finding that the *P. protegens* PBL3 secretome exhibited antimicrobial activity against the rice pathogen *B. glumae* (13, 14) prompted us to evaluate its activity against several plant pathogenic bacteria, such as other *Burkholderia* species, as well as *Xanthomonas campestris* pv. malvacearum, *Erwinia amylovora,* and *Xanthomonas axonopodis*. The antimicrobial assays revealed that the *P. protegens* PBL3 secretome significantly inhibited the growth of all the strains belonging to the genus *Burkholderia,* but it was ineffective

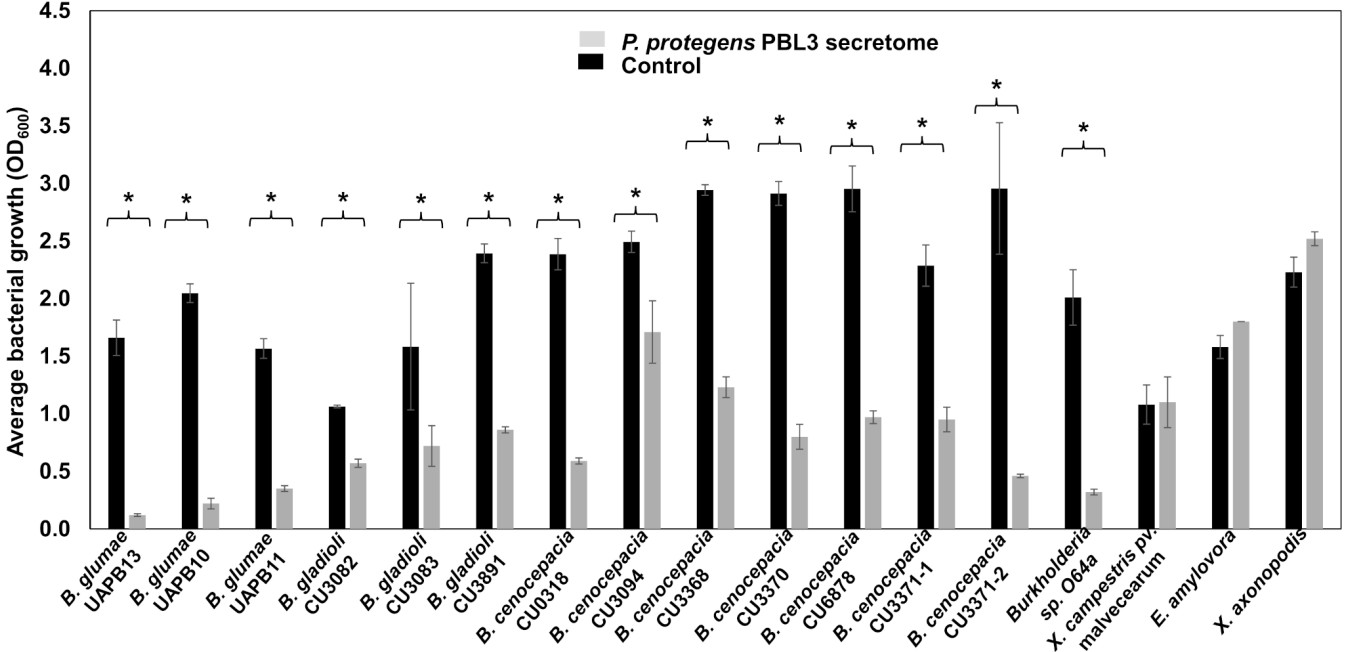

**FIG 1** The *P. protegens* PBL3 secretome inhibits the growth of a broad range of plant pathogenic bacteria. Bacterial strains $OD_{600} = 0.2$ were grown on King's B (KB) broth (black bars) or KB amended with the *P. protegens* PBL3 secretome (gray bars) and incubated at 28°C with constant agitation. Bacterial growth ($OD_{600}$) was measured after 18 h. Bars represent the average bacterial growth, and the line graph above the bar represents the standard deviation of bacterial growth for three replications. Students' *t*-test was used to evaluate statistical significance when comparing bacterial growth in non-amended versus amended media. The asterisk above comparing bars indicates statistically significant differences with a *P* value = 0.05.

against *X. campestris* pv malvacearum, *E. amylovora,* and *X. axonopodis* (Fig. 1; Table S1). The levels of inhibition among *Burkholderia* spp. ranged from 31% to 93% ($P < 0.05$). *B. glumae* strains grown with the *P. protegens* PBL3 secretome showed growth reductions ranging from 78% to 93% compared to their growth without the *P. protegens* PBL3 secretome ($P < 0.05$). The inhibitory effect of *P. protegens* PBL3 secretome against *Burkholderia gladioli* was moderate, with a 46% growth reduction in *B. gladioli* CU3082 ($P < 0.05$), a 54% growth reduction in *B. gladioli* CU3083, and a 64% growth reduction in *B. gladioli* CU3891 ($P < 0.05$). Strains of *Burkholderia cenocepacia* exhibited variable susceptibility against the *P. protegens* PBL3 secretome, with 85% reduction in *B. cenocepacia* CU3371-2 ($P < 0.05$), a reduction of 75% in *B. cenocepacia* 0318 ($P < 0.05$), 67% reduction in *B. cenocepacia* CU6878 ($P < 0.05$), a reduction of 59% in *B. cenocepacia* CU3371-1 ($P < 0.05$) and *B. cenocepacia* 3368 ($P < 0.05$), and a minimal reduction of 31% in *B. cenocepacia* CU3094 ($P < 0.05$). The *Burkholdera* spp. O64a showed a growth reduction of 84% compared with the controls ($P < 0.05$) (Fig. 1). Overall, the *P. protegens* PBL3 secretome displayed broad yet selective antimicrobial activity, showing strong inhibition of *B. glumae* strains and variable suppression of other *Burkholderia* species but no measurable effect on *Xanthomonas* or *Erwinia* strains.

## The *P. protegens* PBL3 secretome demonstrated an inhibitory effect on human pathogens

To evaluate the antimicrobial activity of the *P. protegens* PBL3 secretome against human bacterial pathogens, we selected seven Gram-negative bacteria, including the human pathogens *A. baumannii, E. coli* O157:H7, *Pseudomonas aeruginosa* PAO1, *P. aeruginosa* PA 103, *Salmonella typhi, Y. enterocolitica,* as well as the non-pathogenic laboratory strain *E. coli* K12. We also selected three Gram-positive pathogenic bacteria: *E. faecium, L. innocua, and S. aureus*. The results showed that the *P. protegens* PBL3 had differential activity against Gram-negative bacteria, inhibiting the growth of *A. baumannii, E. coli* O157:H7, and *Y. enterocolitica* (Fig. 2; Table S2). In *A. baumannii,* the *P. protegens* PBL3 secretome at 20% and 30% (vol/vol) caused a 93% reduction in growth ($P < 0.05$) when compared against the control (Fig. 2; Fig. S1A). Similarly, *E. coli* O157:H7 showed an average reduction of 40% ($P < 0.05$) when growth media was amended with *P. protegens* PBL3 secretome at 20% and 30% (vol/vol) in comparison with growth in non-amended media (Fig. 2; Fig. S1B). Similarly, *P. protegens* PBL3 secretome caused a 96% reduction in growth ($P < 0.05$) in *Y. enterocolitica* when the *P. protegens* PBL3 were added at 20% and 30% (vol/vol) (Fig. 2; Fig. S1F). The *P. protegens* PBL3 secretome did not have any effect against the pathogens *P. aeruginosa* PAO1 and PA13, and *S. typhi* or the laboratory strain *E. coli* K12 (Fig. 2).

We also found that the *P. protegens* PBL3 secretome caused a growth reduction in *E. faecium, L. innocua,* and *S. aureus* (Fig. 2; Table S2). In *E. faecium*, the *P. protegens* PBL3 secretome at 20% and 30% (vol/vol) caused a growth reduction of 25% and 55%, respectively ($P < 0.05$), when compared with the controls (Fig. 2; Fig. S2A), whereas in *L. innocua* amended with *P. protegens* PBL3 secretome, a growth reduction of 33% was only observed at 30% (vol/vol) ($P < 0.05$) (Fig. S2B). Similarly, *S. aureus* showed an average reduction of 30% ($P < 0.05$) when the *P. protegens* PBL3 secretome was added at 20% and 30% (vol/vol) (Fig. 2; Fig. S2C). Altogether, these results show that the *P. protegens* PBL3 secretome has a differential level of activity against the Gram-negative pathogens tested, showing higher inhibition of *A. baumannii* and *Y. enterocolitica,* and measurable inhibitory effects on *E. coli* O157:H7, *S. aureus*, *L. innocua*, and *E. faecium*, while having no detectable effect on *E. coli* K12, *P. aeruginosa* PAO1, *P. aeruginosa* PA13, or *S. typhi*.

## DISCUSSION

Our previous findings that *P. protegens* PBL3 has antagonistic activity against *B. glumae* and that such activity was associated with the *P. protegens* PBL3 secretome created an opportunity to harness the *P. protegens* PBL3 secretome to develop effective biopesticides against *B. glumae*. In the past, *B. glumae* infections in rice were treated

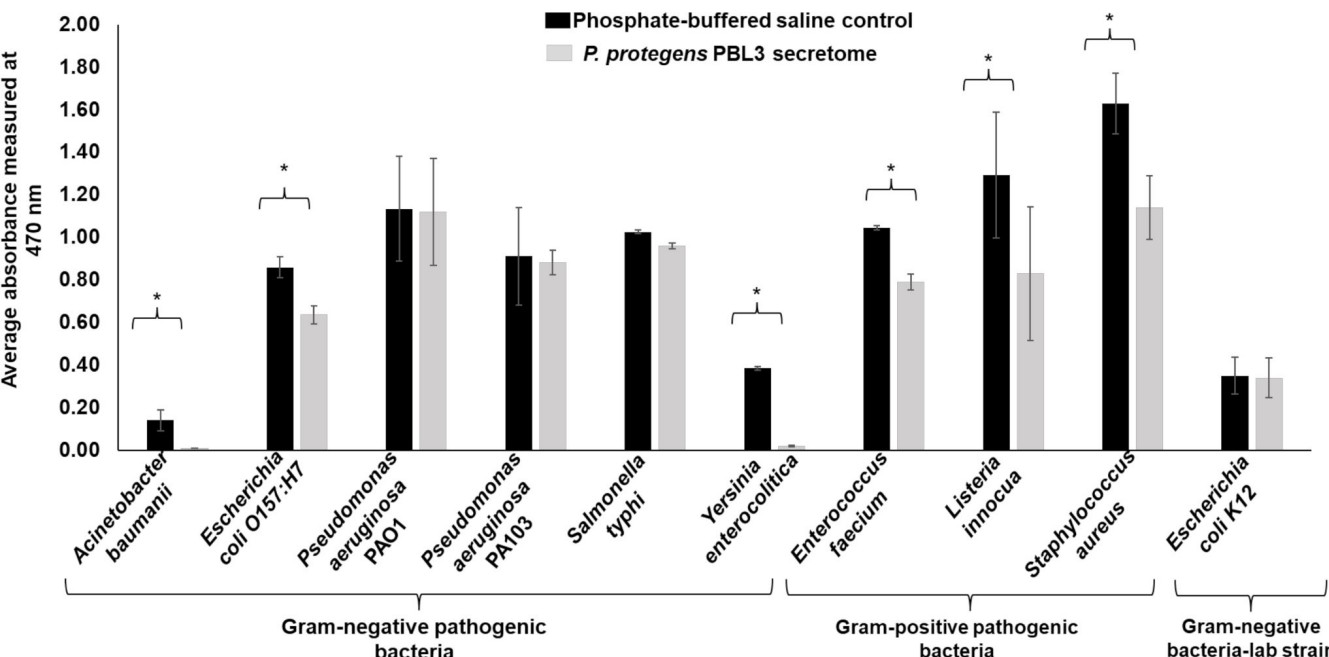

**FIG 2** The *P. protegens* PBL3 secretome inhibits the growth of a broad range of human pathogens. Pathogenic Gram-negative and Gram-positive bacteria and the laboratory strain *E. coli* K12 were grown in agar plates for 20 h. Bacteria were scraped from a plate and diluted in PBS buffer to an optical density 600 ($OD_{600}$) of 0.2 and further diluted in growing media to a final concentration of $1.5 \times 10^6$ CFU/mL. Twenty microliters of the *P. protegens* PBL3 secretome or PBS were mixed with different volumes of growing media to a final volume of 100 µL and grown for 18 h at 37°C with constant agitation. After 18 h, 0.1% 2,3,5-triphenyl-tetrazolium chloride was added to the cultures. Absorbance at 470 nm was read in a plate reader. The bar represents the average absorbance of bacteria minus the absorbance of media alone, and the error bar represents the standard deviation of six replicates. Asterisk suggests a difference between the PBS and PBL3 secretome, with a level of significance set at $P = 0.05$.

with kasugamycin and oxolinic acid, but they proved ineffective with the emergence of resistance (20, 21), prompting research into alternative strategies to control this pathogen (22–24).

In this work, we demonstrated that in addition to *B. glumae,* the *P. protegens* PBL3 secretome has a broad yet selective spectrum of activity inhibiting the growth of other plant pathogenic *Burkholderia* strains, including *B. gladioli,* which also causes BPB, and *B. cenocepacia,* causing sour skin disease in onion (19). While our study focused on phytopathogenic strains of *Burkholderia,* it is important to note that other species within this genus are significant human pathogens. The *Burkholderia cepacia* complex (BCC), which includes at least 24 species, is known to cause opportunistic infections, such as cystic fibrosis-related lung disease, bloodstream infections, and respiratory tract infections (25, 26). Although *B. gladioli* is not part of the BCC, it has also been isolated from patients with cystic fibrosis and is considered clinically relevant (27). The BCC group is notoriously difficult to treat due to both intrinsic and acquired resistance to multiple classes of antimicrobials, leaving few effective treatment options (25). While we did not test any BCC-associated human pathogens in this study, the demonstrated antimicrobial activity of the *P. protegens* PBL3 secretome against *B. gladioli* and other *Burkholderia* spp. suggests its potential relevance beyond plant disease management and warrants further exploration in the context of human health.

In that regard, we found that the *P. protegens* PBL3 secretome was also effective against human pathogens, such as *A. baumannii*, *E. coli* O157:H7, *Y. enterocolitica*, *S. aureus*, and *E. faecium*. The activity of the *P. protegens* PBL3 secretome against the human pathogens tested is significant in the context of AMR. *A. baumannii*, listed by the World Health Organization in 2017 as a top-priority (critical) pathogen for the development of new antimicrobials, is often multidrug-resistant and is associated with mortality rates of up to 60% in hospital-acquired pneumonia and bloodstream infections (28–30). The *A.*

*baumannii* strain used in our study is an antimicrobial-resistant clinical isolate recently obtained from a patient.

Similarly, the WHO Tier 2 high-priority list includes other clinically significant bacteria, such as *E. faecium* and *S. aureus*, which are also increasingly resistant to current antibiotics (30). While our study did not specifically evaluate antibiotic-resistant strains of these pathogens, we observed measurable inhibition of their growth when exposed to the PBL3 secretome. These findings suggest the presence of compounds with activity against a broad range of pathogens and support the potential of this secretome as a source for future antimicrobial discovery.

Our previous results, validating genomic predictions by evaluating the antimicrobial activity against *B. glumae* of eight purified secondary metabolites: Orfamide A, Orfamide B, Fengycin, Pyrroniltrin, Pyochelin, Pyoverdine, 2,4 DAPG, and Pyoluteroin revealed that only Pyoverdine, 2,4 DAPG, and Pyoluteroin had antimicrobial activity at high concentrations. However, those concentrations were not equivalent to their respective endogenous concentrations in the *P. protegens* PBL3 secretome, implying that the overall antimicrobial activity of the *P. protegens* PBL3 secretome is associated with novel molecules or a combination of molecules (14).

Although the specific molecules associated with the antimicrobial activity in the *P. protegens* PBL3 secretome remain to be identified, some of the *P. protegens* PBL3 genome-predicted secondary metabolites have been previously studied. The exposure of bacterial cells to 2,4-DAPG leads to cell membrane damage and cell content disintegration (31, 32). Pyoverdine is a high-affinity siderophore whose antimicrobial properties are associated with ferric iron ($Fe^{3+}$) sequestration that deprives pathogen cells of iron (33, 34). Pyoluteorin damages the bacterial cell membrane, causing cytoplasmic leakage, as well as inducing the accumulation of reactive oxygen species, leading to lethal oxidative damage (35). These multiple modes of action associated with molecules expected to be in the *P. protegens* PBL3 secretome suggest that the *P. protegens* PBL3 secretome could have a multifaceted effect, simultaneously targeting several bacterial processes.

Despite the broad spectrum of activity of the *P. protegens* PBL3 secretome, our results also showed differential activity among strains, suggesting that bioactive molecule(s) in the *P. protegens* PBL3 secretome have unique target(s) that are present in *P. protegens* PBL3-sensitive bacteria and absent in the *P. protegens* PBL3-insensitive bacteria. Hypothetically, bacteria that are insensitive to the *P. protegens* PBL3 secretome employ mechanisms known to drive AMR, such as detoxification, efflux pumps, or specific adaptive strategies to prevent internalization of bioactive molecules (36). Our results also revealed that *P. aeruginosa* and *S. typhi* were insensitive to the *P. protegens* PBL3 secretome. *P. aeruginosa* has intrinsic resistance to antimicrobials through several mechanisms, including (i) unique outer membrane composition that prevents antibiotic uptake, (ii) downregulation of membrane porins required for antimicrobial uptake, (iii) presence of multiple efflux pump systems which actively expel a wide range of structurally unrelated antibiotics, (iv) constitutive production of antibiotic hydrolyzing enzymes, and (v) biofilm formation that provides a protective shield restricting antibiotic penetration and promotes nutrient and oxygen gradients (37, 38). The resistance mechanisms of *S. typhi* against frontline antibiotics, particularly fluoroquinolones and third-generation cephalosporins, are complex and often plasmid-mediated. *S. typhi* exhibits antibiotic resistance through multiple mechanisms, including (i) chromosomal mutations which alter the drug target, (ii) plasmid-mediated mechanisms that protect target enzymes and decrease intracellular drug accumulation, (iii) production of antibiotic hydrolyzing enzymes, (iv) overexpression of efflux pumps that actively expel antibiotics from the cell, and (v) decreased outer membrane permeability resulting from porin modifications (39, 40).

The broad-spectrum antimicrobial profile of the *P. protegens* PBL3 secretome is significant, given that the affected pathogens are usually sensitive to antibiotics representing multiple mechanistic classes. For instance, β-lactam antibiotics, which

act by binding to penicillin-binding proteins involved in peptidoglycan biosynthesis, are commonly used to treat infections caused by *A. baumannii, S. aureus,* and *L. innocua* (28, 41). Likewise, *Y. enterocolitica* is susceptible to aminoglycosides, which exert their effects by irreversibly binding to the 30S ribosomal subunit and inhibiting protein synthesis, while fluoroquinolones inhibit bacterial DNA gyrase and topoisomerase IV, preventing DNA replication, used for the treatment of *S. typhi* infections (39). The inhibition of multiple bacterial genera by the *P. protegens* PBL3 secretome, each typically sensitive to distinct classes of antibiotics, highlights its potential to contain a combination of antimicrobial compounds that act through different or overlapping molecular targets.

In clinical settings, numerous antibiotic classes are available for the treatment of human bacterial infections; however, in agriculture, only three antibiotics, streptomycin, kasugamycin, and oxytetracycline, are currently registered in the United States for the management of bacterial plant diseases (7). Consequently, the broad-spectrum antimicrobial activity exhibited by the *P. protegens* PBL3 secretome against multiple plant pathogens highlights its potential as a novel biological alternative for sustainable disease management. This is particularly relevant in agricultural systems where the limited availability of effective antimicrobial agents, coupled with the growing concern over resistance development, highlights the urgent need for safe and environmentally compatible biocontrol solutions.

Overall, our study shows that the *P. protegens* PBL3 secretome contains bioactive molecules with activity against diverse plant and human bacterial pathogens that could be used as a source of antimicrobial compounds for the development of novel therapeutics. Although the specific compounds remain to be identified, ongoing work aims to identify and validate them.

## ACKNOWLEDGMENTS

We would like to thank Dr. Martin Conda Sheridan (University of Nebraska Medical Center) and Dr. Byron Chaves (Department of Food Science and Technology, University of Nebraska-Lincoln) for sharing protocols and bacterial strains, and to Dinithi De Silva for her valuable assistance during the experiments.

## AUTHOR AFFILIATIONS

[1]Department of Plant Pathology, University of Nebraska-Lincoln, Lincoln, Nebraska, USA
[2]Complex Biosystem Program, University of Nebraska-Lincoln, Lincoln, Nebraska, USA
[3]University of Nebraska-Lincoln, Nebraska Centre for Biotechnology, Lincoln, Nebraska, USA
[4]University of Nebraska-Lincoln, Center for Plant Science Innovation, Lincoln, Nebraska, USA

## AUTHOR ORCIDs

Clemencia M. Rojas (iD) http://orcid.org/0009-0004-4269-5768

## FUNDING

| Funder | Grant(s) | Author(s) |
| --- | --- | --- |
| Nebraska Research Initiative | | Clemencia M. Rojas |

## ADDITIONAL FILES

The following material is available online.

## Supplemental Material

**Supplemental material (Spectrum02669-25-s0001.pdf).** Tables S1 and S2; Fig. S1 and S2.

## Open Peer Review

**PEER REVIEW HISTORY (review-history.pdf).** An accounting of the reviewer comments and feedback.

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
