## [Reviewer comments · Microbiology Spectrum]

Microbiology Spectrum

The secretome of the environmental bacterium *Pseudomonas protegens* PBL3 has broad-spectrum antimicrobial activity against plant and human pathogenic bacteria

Biwesh Ojha, Kiyara Grosz, and Clemencia Rojas

Corresponding Author(s): Clemencia Rojas, University of Nebraska-Lincoln

Review Timeline:

Submission Date:	September 1, 2025
Editorial Decision:	October 2, 2025
Revision Received:	October 24, 2025
Accepted:	October 27, 2025

Editor: Lindsey Burbank

Reviewer(s): Disclosure of reviewer identity is with reference to reviewer comments included in decision letter(s). The following individuals involved in review of your submission have agreed to reveal their identity: Liezel del Castillo Atole (Reviewer #1)

Transaction Report:

DOI: <https://doi.org/10.1128/spectrum.02669-25>

Re: Spectrum02669-25 (The environmental bacterium *Pseudomonas protegens* PBL3 has broad-spectrum antimicrobial activity against plant and human pathogenic bacteria)

Dear Dr. Clemencia M. Rojas:

Thank you for the privilege of reviewing your work. Below and in the attached file, you will find instructions from the Spectrum editorial office, and the reviewer comments for modification of your manuscript. Please address all reviewer comments in your resubmission.

Revision Guidelines

Sincerely,
Lindsey Burbank
Editor
Microbiology Spectrum

Reviewer #1 (Comments for the Author):

The topic is highly relevant given the urgent need for new antimicrobial compounds in agriculture and clinical settings. The study is well-designed, and the results are presented clearly. The secretome's broad yet selective antimicrobial activity against several important pathogens is a significant finding.

Reviewer #2 (Comments for the Author):

1. It is suggested to clarify the role of "secretome" in the title, for example: "The secretome of *Pseudomonas protegens* PBL3 exhibits broad-spectrum antimicrobial activity against plant and human pathogenic bacteria"
2. The abstract does not mention specific inhibition rates or names of key pathogens. It is recommended to add 1-2 representative results (e.g., 95% inhibition against *A. baumannii*).
3. The introduction section can strengthen the research background: the introduction of *P. protegens* and its secondary metabolites can be more systematic, especially its relationship with known antibiotics.
4. The discussion section can strengthen the speculation on mechanisms, although the compounds have not been identified, it is still possible to speculate on their potential targets by combining the known mechanisms of known metabolites (such as pyoverdine, DAPG).
5. The specific composition of "Minimal 9 media" is not stated in the text. It is recommended to supplement it to facilitate the repetition of the experiment.
6. It is not clear whether a "bacteria-free secretome control" has been set up to rule out the impact of the secretome itself on OD600.
7. The text uses "%v/v"; it is recommended to unify it as "% (v/v)" or clarify whether it refers to the final concentration.
8. The article does not explain why TTC staining was chosen and its linear relationship with the number of viable bacteria. It is recommended to add a brief explanation of the principle.
9. Only "student's t-test" is mentioned, without specifying whether it is two-tailed, paired, or unpaired, and no specific p-value is provided.
10. Lack of representative strain names in Figure 1. The specific *Burkholderia* strain names are not labeled in the bar chart. It is recommended to indicate them in the figure or figure caption.
11. The label "Gram-positive bacteria" in Figure 2 is incorrect, the figure contains Gram-negative bacteria (such as *A. baumannii*). It is recommended to correct the figure caption or reclassify them.
12. The expression of inhibition rate data is inconsistent, for example, "~30%", ">30%", "~95%". It is recommended to unify it into the form of "mean {plus minus} standard deviation".
13. Supplementary Figure 1-2 lacks significance markers, although the text mentions "Asterisk suggests a difference", the asterisk (*) is not shown in the figure and needs to be supplemented.
14. Insufficient explanation of ineffective strains. For example, there is no inhibition of *P. aeruginosa* and *S. typhi*, and the possible mechanisms (such as efflux pumps, biofilms, etc.) are not discussed.
15. Not compared with known antibiotics. The lack of comparison of activity with antibiotics commonly used in current clinical or agricultural settings makes it difficult to evaluate their practical application potential.
16. Inconsistent use of italics for strain names. For example, *Burkholderia* is sometimes not italicized; it is recommended to unify this throughout the text.
17. The format of the references needs to be unified, page numbers are missing in some references (e.g., Sati et al., 2025), and the DOI format is incomplete in others.

**The environmental bacterium *Pseudomonas protegens* PBL3 has broad-spectrum**
**antimicrobial activity against plant and human pathogenic bacteria**

Biwesh Ojha^{1,2}, Kiyara Grosz³, Clemencia M. Rojas^{1,4*}

¹ Department of Plant Pathology, University of Nebraska-Lincoln

² Complex Biosystem Program, University of Nebraska-Lincoln

³ Nebraska Centre for Biotechnology, University of Nebraska-Lincoln

⁴ Center for Plant Science Innovation, University of Nebraska-Lincoln

* Corresponding Author: Clemencia M. Rojas, crojas2@unl.edu

**Abstract**

Antimicrobials are widely used to manage bacterial diseases in humans, pets, and
livestock. Unfortunately, their extensive use has contributed to the emergence of
antimicrobial resistance (AMR), a threat to global health. In contrast to the widespread
antimicrobial use in medicine and veterinary, the use of antimicrobials to control bacterial
diseases in plants is limited, posing a threat to global food security. These two threats
highlight the urgent need to discover novel antimicrobial compounds with new modes of
action. Previously, we demonstrated that the environmental bacterium *Pseudomonas*
*protegens* PBL3 inhibits the growth of the plant pathogenic bacterium *Burkholderia*
*glumae*, indicating that this strain produces and secretes potent antimicrobial compounds.
In this study, we assessed the antimicrobial activity of the *P. protegens* PBL3 secretome
against selected plants and human bacterial pathogens and discovered its broad
spectrum of activity. These results highlight the *P. protegens* PBL3 secretome as a
promising source of natural antimicrobial compounds with potential applications in both
agriculture and clinical settings.

**Importance**

Bacterial diseases threaten both human health and global food production. While
antimicrobials have been vital for saving lives and protecting animals, their overuse has
led to antimicrobial resistance (AMR), where bacteria no longer respond to available
treatments. This has created an urgent need for new and safer ways to control harmful
bacteria. In agriculture, the problem is especially severe because very few antimicrobials
are approved for use on crops, leaving farmers with limited tools to manage devastating

plant diseases. Our study focuses on *Pseudomonas protegens* PBL3, a naturally
occurring soil bacterium that produces and releases antimicrobial molecules. We found
that its secretome, the mixture of compounds it secretes, can inhibit the growth of a wide
range of pathogenic bacteria, including those that cause crop loss and serious human
infections. These findings highlight the potential of beneficial microbes as a sustainable
source of advancing the development of antimicrobials for both agriculture and medicine.

**Introduction**

Antimicrobials to control bacterial diseases have long been used to manage bacterial
infections in humans, pets, livestock, and aquatic animals as therapeutic or prophylactic
agents (Prescott, 2017; Schar et al., 2020; Van Boeckel et al., 2017; Velazquez-Meza et
al., 2022). However, this widespread use of antimicrobials has significantly contributed to
the emergence of antimicrobial resistance (AMR) (Velazquez-Meza et al., 2022). The
AMR threat is becoming a matter of global concern that has not been mitigated, as most
of the recently approved antimicrobials are derivatives of previously known classes
(Miethke et al., 2021; WHO, 2023), and bacteria quickly evolve mechanisms to render
them ineffective. In contrast to the widespread use of antimicrobials in the medical and
veterinary fields, the use of antimicrobials to combat bacterial diseases in crops has been
limited, as effective antimicrobials for those pathogens have not been identified, have
proven ineffective, or only approved for agricultural use for a limited number of bacterial
pathogens (Sundin & Wang, 2018; Verhaegen et al., 2023). Thus, both the threat of
antimicrobial resistance in the medical and veterinary fields and the absence of effective
antimicrobials in agriculture necessitate the discovery of new molecules with unique
chemistries and modes of action.

Historically, naturally occurring microorganisms have been a prolific source of
antimicrobial compounds since the discovery of penicillin from the fungus *Penicillium*
*notatum* that marked the beginning of the antimicrobial era and demonstrated the
therapeutic potential of microbial metabolites (Demain & Sanchez, 2009). Since then,
various fungal species have yielded diverse classes of antimicrobials. Beyond fungi,
bacteria, particularly actinomycetes, have played a dominant role in antimicrobial

discovery. Of the estimated 22,500 biologically active microbial compounds, more than
64% of all known antimicrobial classes are derived from the genus *Streptomyces* (Berdy,
2005; Hutchings et al., 2019). Additional contributors of antimicrobials are bacteria from
the genera *Bacillus*, *Paenibacillus*, and *Pseudomonas* (Hutchings et al., 2019). Although
natural products from microorganisms have yielded the majority of clinically used
antimicrobials, a vast reservoir of untapped bioactive compounds still exists in microbial
genomes.

Previously, the environmental bacterium *Pseudomonas protegens* PBL3 was identified
as an antagonistic bacterium to the plant pathogenic bacterium *Burkholderia glumae*, the
causal agent of the rice disease Bacterial Panicle Blight (Ortega & Rojas, 2021). The
antagonistic activity of *P. protegens* PBL3 against *B. glumae* was associated with the *P.*
*protegens* PBL3 secreted fraction (secretome), indicating that *P. protegens* PBL3
produces and secretes antimicrobials (Ortega et al., 2020). Those antimicrobials are likely
associated with the multiple genetic clusters in the *P. protegens* PBL3 genome predicted
to at least 14 biosynthetic clusters predicted to encode secondary metabolites (Ortega et
al., 2020). Only eight of those were commercially available for testing, and out of those
eight, only pyoverdine, pyoluteorin and 2,4-diacetylphloroglucinol demonstrated
antimicrobial activity, inhibiting the growth of *B. glumae* at high concentrations, yet not
fully explaining the antimicrobial activity of *P. protegens* PBL3 secretome (Dahal et al.,
2024). Although the complete chemical composition of the *P. protegens* PBL3 secretome
is still unknown, the predicted complexity based on its coding capacity led us to
hypothesize that the *P. protegens* PBL3 secretome possesses broad-spectrum
antimicrobial activity against other bacterial pathogens. In this study, we investigated the

spectrum of activity of the *P. protegens* PBL3 secretome against plant and human
pathogenic bacteria. Our results showed that the secretome inhibited the growth of plant
pathogenic bacteria of the genera *Burkholderia* and *Xanthomonas*, as well as selected
human pathogens such as *Acinetobacter baumannii*, *Escherichia coli* O157:H7, *Listeria*
*innocua*, *Salmonella typhi*, *Enterococcus faecium*, and *Staphylococcus aureus*. These
findings suggest that the *P. protegens* PBL3 secretome represents a source of
antimicrobials of broad-spectrum and possibly novel mechanisms of action for the
management of both plant and human bacterial pathogens.

**Materials and Methods**

**Bacterial strains**

Bacterial strains in this study are listed in Table 1. *P. protegens* PBL3 were grown on LB
agar, plant pathogens were grown on KB agar, and human pathogens were grown on
Tryptic Soy agar. Bacterial strains were streaked from -80 °C freezer on agar plates and
incubated at 28°C for plant pathogenic bacteria and 37°C for human pathogenic bacteria.

**Antimicrobial assay against plant pathogenic bacteria**

A single colony of *P. protegens* PBL3 was grown in Minimal 9 media for 36-48 hours until
the Optical density measured at 600nm (OD₆₀₀) reached 0.5. The secreted fraction
(secretome) was separated from whole bacterial cells by centrifugation, and filter-
sterilized as previously described (Dahal et al., 2024).

Plant pathogenic bacterial strains were scraped from the agar plates and resuspended in
sterile water to a final concentration of OD₆₀₀ =0.2. The bacterial suspension was mixed
with either *P. protegens* PBL3 secretome or water (control) and incubated in a shaker for

18h. Bacterial growth was evaluated by measuring the OD₆₀₀. Each treatment had three
replications and was repeated three times.

**Antimicrobial assay against human pathogenic bacteria**

Human pathogenic bacterial strains were scraped from the agar plates and resuspended
in 1X Phosphate-Buffered Saline (PBS) to an OD₆₀₀ of 0.2 (1.5×10^8 colony-forming units
(CFU)/mL) and further diluted in a growing media to a final bacterial concentration of 1.5
$\times 10^6$ CFU/mL. The antimicrobial activity of the *P. protegens* PBL3 secretome was
assessed in a 96-well microtiter plate. Each well contained a total volume of 100 μ L,
consisting of growth media, *P. protegens* PBL3 secretome (or PBS control), and the
bacterial suspension. A fixed volume of 10 μ L of the bacterial suspension to be evaluated
was added to both treatment and control wells. The antimicrobial potency of *P. protegens*
PBL3 was evaluated using a microdilution assay, where varying volumes (10, 20, and 30
μ L) of the PBL3 secretome were tested, adjusting the final volume of 100 μ L with growth
media. Each treatment was conducted in six replicates. For the controls, PBS was added
in volumes equivalent to those of the secretome. Growth media alone (100 μ L) was used
as negative control, while 10 μ L of bacterial suspension in 90 μ L of growth media was
used as positive control. Microtiter plates were incubated at 37°C for 18 hours. For visual
confirmation of bacterial growth, 2,3,5-Triphenyl-tetrazolium chloride (TTC) solution
(0.1%) was added to each well and incubated at 37°C for 30 min. The bacterial growth
was quantified by measuring the TTC color change at an absorbance of 470 nm (A_{470nm})
using a plate reader (Synergy™ 2 Multi-mode microplate reader, Biotek® Instruments,
Inc.)

**Statistical Analysis**

Data analysis was conducted using R Studio version 4.2.1, calculating the mean and
standard deviation of bacterial growth measured for both *P. protegens* PBL3 treatment
and control groups in plant and human pathogens. A student's t-test was performed to
determine differences between the control and *P. protegens* PBL3 secretome, with the
significance level set at $p=0.05$.

**Results**

**The *P. protegens* PBL3 secretome has a broad-spectrum activity against plant** 139 **pathogens**

The finding that the *P. protegens* PBL3 secretome exhibited antimicrobial activity against
the rice pathogen *Burkholderia glumae* (Dahal et al., 2024; Ortega et al., 2020), prompted
142 us to evaluate its activity against several plant pathogenic bacteria, such as other
*Burkholderia* species, as well as *Xanthomonas campestris* pv. malvacearum, *Erwinia*
*amylovora*, and *Xanthomonas axonopodis*. The antimicrobial assays revealed that the *P.*
*protegens* PBL3 secretome significantly inhibited the growth of all the strains belonging
to the genus *Burkholderia*, but it was ineffective against *X. campestris* pv malvacearum,
*E. amylovora* and *X. axonopodis* (Figure 1). The levels of inhibition among *Burkholderia*
spp ranged from 31% to 93%. *Burkholderia glumae* strains grown with the *P. protegens*
PBL3 secretome showed growth reductions ranging from 78% to 93% compared to their
growth without the *P. protegens* PBL3 secretome. The inhibitory effect of *P. protegens*
PBL3 secretome against *B. gladioli* was moderate, with a 46% growth reduction in *B.*
*gladioli* CU3082, a 54% growth reduction in *B. gladioli* CU3083, and a 64% growth
reduction in *B. gladioli* CU3891. Strains of *B. cenocepacia* exhibited variable susceptibility

against the *P. protegens* PBL3 secretome, with 85% reduction in *B. cenocepacia*
CU3371-2, a reduction of 75% in *B. cenocepacia* 0318, 67% reduction in *B. cenocepacia*
CU6878, a reduction of 59% in *B. cenocepacia* CU3371-1 and *B. cenocepacia* 3368, and
a minimal reduction of 31% in *B. cenocepacia* CU3094. The *Burkholderia* spp O64a
showed a growth reduction of 84% compared with the controls (Figure 1). Overall, the *P.*
*protegens* PBL3 secretome displayed broad yet selective antimicrobial activity, showing
strong inhibition of *B. glumae* strains and variable suppression of other *Burkholderia*
species but no measurable effect on *Xanthomonas* or *Erwinia* strains.

**The *P. protegens* PBL3 secretome demonstrated an inhibitory effect on human** 164 **pathogens**

To evaluate the antimicrobial activity of the *P. protegens* PBL3 secretome against human
bacterial pathogens, we selected 6 Gram-negative pathogenic bacteria: *Acinetobacter*
*baumannii*, *Escherichia coli* O157:H7, *Pseudomonas aeruginosa* PAO1, *Pseudomonas*
*aeruginosa* PA 13, *Salmonella typhi*, *Yersinia enterocolitica* and the laboratory strain
*Escherichia coli* K12. We also selected 3 Gram-positive pathogenic bacteria:
*Enterococcus faecium*, *Listeria innocua*, and *Staphylococcus aureus*. The results showed
that the *P. protegens* PBL3 had differential activity against gram-negative bacteria.
inhibiting the growth of *A. baumannii*, *E. coli* O157:H7, and *Y. enterocolitica*. In *A.*
*baumannii*, the *P. protegens* PBL3 secretome at 20 and 30 v/v% caused a 95% reduction
in growth when compared against the control (Figure 2 and Supplementary Figure 1A).
Similarly, *Escherichia coli* O157:H7 showed a reduction of ~ 40% when growth media was
amended with *P. protegens* PBL3 secretome at 20 and 30 %v/v in comparison with growth

in non-amended media (Figure 2 and Supplementary Figure 1B). Similarly, *P. protegens*
PBL3 secretome caused a reduction of ~95% growth in *Y. enterocolitica* when the *P.*
*protegens* PBL3 were added at 20 and 30%v/v (Figure 2 and Supplementary Figure 1F).
The *P. protegens* PBL3 secretome did not have any effect against the pathogens *P.*
*aeruginosa* PAO1 and PA13, and *S. typhi* or the laboratory strain *E. coli* K12 (Figure 2).
We also found that the *P. protegens* PBL3 secretome inhibits the growth of *E. faecium*, *L.*
*innocua*, and *S. aureus* (Figure 2). The *P. protegens* PBL3 secretome at 20 and 30%v/v
caused a reduction of ~30% growth of *E. faecium* when compared with the controls
(Figure 2 and Supplementary Figure 2A), whereas *L. innocua* amended with *P. protegens*
PBL3 secretome only exhibited its effect at 30%v/v and caused a reduction of growth by
55% (Supplementary Figure 2B). Similarly, *S. aureus* showed an average reduction of
more than 30% when the *P. protegens* PBL3 secretome were added at 20 and 30%v/v
(Figure 2 and Supplementary Figure 2C). The *P. protegens* PBL3 secretome
demonstrates a differential level of activity against the gram-negative pathogens, showing
higher inhibition of *A. baumannii* and *Y. enterocolitica*, and measurable inhibitory effects
on *E. coli* O157:H7, *S. aureus*, *L. innocua*, and *E. faecium*, while having no detectable
effect on *E. coli* K12, *P. aeruginosa* PAO1, *P. aeruginosa* PA13, or *Salmonella typhi*.

**Discussion**

In this work, we demonstrated that the *P. protegens* PBL3 secretome has a broad yet
selective spectrum of activity inhibiting the growth of several plant pathogenic
*Burkholderia* strains as well as strains from the human pathogens *A. baumannii*, *E. coli*
O157:H7, *Y. enterocolitica*, *S. aureus*, and *E. faecium*. The specificity of the antimicrobial

activity against some of the bacterial strains tested, but not all, suggests that specific
bioactive molecules in the *P. protegens* PBL3 secretome have specific targets on the
bacteria tested.

Previous work from our lab, uncovered at least 14 biosynthetic clusters predicted to
encode the secondary metabolites: two different types of Pyoverdines, Pyrrolnitrin,
Pyoluteorin, 2,4-diacetylphloroglucinol, Orfamide C, Arylpolyene, 2-
Hydroxyphenylthiazoline, thiazostatin, watasemycin B, Fengycin, Lipopeptide 8D1-1, 2
Bacteriocin, Pyochelin, Cyclodipeptides and N-acetylglutamine amide (Ortega et al.,
2020). Several of those secondary metabolites such as: 2,4-diacetylphloroglucinol,
thiazostatin, and fengycin A have demonstrated antifungal, but not antibacterial activity
(Gimenez et al., 2021; Marchand et al., 2000; Sasaki et al., 2002; Umemura et al., 2021);
only watasemycin B, have demonstrated, yet very weak antibacterial activity (Sasaki et
al., 2002). We previously evaluated eight commercially available analogs (Orfamide A and
B, Fengycin, Pyrrolnitrin, Pyochelin, Pyoverdine, 2,4 DAPG, Pyoluteroin) against *B.*
*glumae*, which revealed that only three (Pyoverdine, 2,4 DAPG, and Pyoluteroin) had
activity at high concentrations, but those concentrations were not equivalent to the
endogenous concentrations in the *P. protegens* PBL3 secretome (Dahal et al., 2024) .
While these results are insightful towards harnessing as potential antimicrobials against
*B. glumae*, their activity at high concentrations, compounded with the high cost of each
of those compounds, makes their potential use impractical. It is not known the
effectiveness of these secondary metabolites against the bacterial pathogens used in this
study.

In the past, *B. glumae* infections in rice were treated with Kasugamycin and Oxolinic acid
but proved ineffective with the emergence of resistance (Maeda et al., 2004, 2007). Few
studies have focused on developing alternate antimicrobials against *B. glumae* (Betancur
et al., 2020; Mohammed-Naji et al., 2025); however, these studies are focused on *in vitro*
with no follow-up studies have been conducted to demonstrate its effectiveness in field
conditions. Thus, the use of the whole *P. protegens* PBL3 secretome could offer a solution
to control BPB (Dahal et al, unpublished).

The results from this research revealed that the *P. protegens* PBL3 secretome was also
effective inhibiting the growth of the other plant pathogenic *Burkholderia* such *B. gladioli*
that also causes BPB and *B. cenocepacia* strains causing sour skin disease in onion
(Asselin et al., 2016). While our study focused on phytopathogenic strains of
*Burkholderia*, it is important to note that other species within this genus are significant
human pathogens. The *Burkholderia cepacia* complex (BCC), which includes at least 24
species, is known to cause opportunistic infections such as cystic fibrosis-related lung
disease, bloodstream infections, and respiratory tract infections (Gutierrez Santana &
Coria Jimenez, 2024; Tavares et al., 2020). Although *B. gladioli* is not part of the BCC, it
has also been isolated from patients with cystic fibrosis and is considered clinically
relevant (Lipuma, 2010). The BCC group is notoriously difficult to treat due to both intrinsic
and acquired resistance to multiple classes of antimicrobials, leaving few effective
treatment options (Gutierrez Santana & Coria Jimenez, 2024). While we did not test any
BCC-associated human pathogens in this study, the demonstrated antimicrobial activity
of the *P. protegens* PBL3 secretome against *B. gladioli* and other *Burkholderia* sp.

suggests its potential relevance beyond plant disease management and warrants further
exploration in the context of human health.

The activity of the *P. protegens* PBL3 secretome against other human pathogens is
significant in the context of antimicrobial resistance (AMR). *Acinetobacter baumannii*,
listed by the World Health Organization in 2017 as a top-priority (critical) pathogen for the
development of new antimicrobials, is often multidrug-resistant, including carbapenem-
resistant strains (CRAB), and is associated with mortality rates of up to 60% in hospital-
acquired pneumonia and bloodstream infections (Peleg et al., 2008; Russo et al., 2019;
Sati et al., 2025). In our study, we tested the *P. protegens* PBL3 secretome against an
antimicrobial-resistant clinical isolate of *A. baumannii* obtained from the UNMC and
observed substantial growth inhibition.

Similarly, the WHO Tier 2 high-priority list includes other clinically significant bacteria,
such as *Enterococcus faecium* and *Staphylococcus aureus*, which are also increasingly
resistant to current antibiotics (Sati et al., 2025). While our study did not specifically
evaluate antibiotic-resistant strains of these pathogens, we observed measurable
inhibition of their growth when exposed to the PBL3 secretome. These findings suggest
the presence of compounds with activity against a broad range of pathogens and support
the potential of this secretome as a source for future antimicrobial discovery.

Overall, our study shows that the *P. protegens* PBL3 secretome contains bioactive
molecules with activity against diverse plant and human bacterial pathogens that could
be used as a source of antimicrobial compounds to the development of novel
therapeutics. Although the specific compounds remain to be identified, ongoing work
aims to identify and validate them.

**Cited references**

- Asselin, J. A. E., Bonasera, J. M. & Beer, S. V. (2016). PCR Primers for Detection of
 *Pantoea ananatis*, *Burkholderia* spp., and *Enterobacter* sp. from Onion. *Plant*
 *Dis*, 100(4), 836-846. <https://doi.org/10.1094/PDIS-08-15-0941-RE>
 Berdy, J. (2005). Bioactive microbial metabolites. *J Antibiot (Tokyo)*, 58(1), 1-26.
 <https://doi.org/10.1038/ja.2005.1>
 Betancur, L. A., Forero, A. M., Vinchira-Villarraga, D. M., Cárdenas, J. D., Romero-
 Otero, A., Chagas, F. O., Pupo, M. T., Castellanos, L. & Ramos, F. A. (2020).
 NMR-based metabolic profiling to follow the production of anti- phytopathogenic
 compounds in the culture of the marine strain sp. PNM-9. *Microbiological*
 *Research*, 239. <https://doi.org/10.1016/j.micres.2020.126507>
 Dahal, S., Alvarez, S., Balboa, S. J., Hicks, L. M. & Rojas, C. M. (2024). Defining the
 Secondary Metabolites in the *Pseudomonas protegens* PBL3 Secretome with
 Antagonistic Activity Against *Burkholderia glumae*. *Phytopathology*, 114(12),
 2481-2490. <https://doi.org/10.1094/PHYTO-04-24-0140-R>
 Demain, A. L. & Sanchez, S. (2009). Microbial drug discovery: 80 years of progress.
 *Journal of Antibiotics*, 62(1), 5-16. <https://doi.org/10.1038/ja.2008.16>
 Gimenez, D., Phelan, A., Murphy, C. D. & Cobb, S. L. (2021). Fengycin A Analogues
 with Enhanced Chemical Stability and Antifungal Properties. *Org Lett*, 23(12),
 4672-4676. <https://doi.org/10.1021/acs.orglett.1c01387>
 Gutierrez Santana, J. C. & Coria Jimenez, V. R. (2024). *Burkholderia cepacia* complex in
 cystic fibrosis: critical gaps in diagnosis and therapy. *Ann Med*, 56(1), 2307503.
 <https://doi.org/10.1080/07853890.2024.2307503>
 Hutchings, M. I., Truman, A. W. & Wilkinson, B. (2019). Antibiotics: past, present and
 future. *Current Opinion in Microbiology*, 51, 72-80.
 <https://doi.org/10.1016/j.mib.2019.10.008>
 Lipuma, J. J. (2010). The changing microbial epidemiology in cystic fibrosis. *Clin*
 *Microbiol Rev*, 23(2), 299-323. <https://doi.org/10.1128/CMR.00068-09>
 Maeda, Y., Kiba, A., Ohnishi, K. & Hikichi, Y. (2004). Implications of amino acid
 substitutions in GyrA at position 83 in terms of oxolinic acid resistance in field
 isolates of *Burkholderia glumae*, a causal agent of bacterial seedling rot and
 grain rot of rice. *Applied and Environmental Microbiology*, 70(9), 5613-5620.
 <https://doi.org/10.1128/Aem.70.9.5613-5620.2004>
 Maeda, Y., Kiba, A., Ohnishi, K. & Hikichi, Y. (2007). Amino acid substitutions in GyrA of
 *Burkholderia glumae* are implicated in not only oxolinic acid resistance but also
 fitness on rice plants. *Appl Environ Microbiol*, 73(4), 1114-1119.
 <https://doi.org/10.1128/AEM.02400-06>
 Marchand, P. A., Weller, D. M. & Bonsall, R. F. (2000). Convenient synthesis of 2,4-
 diacetylphloroglucinol, a natural antibiotic involved in the control of take-all
 disease of wheat. *J Agric Food Chem*, 48(5), 1882-1887.
 <https://doi.org/10.1021/jf9907135>
 Miethke, M., Pieroni, M., Weber, T., Bronstrup, M., Hammann, P., Halby, L., Arimondo, P.,
 B. Glaser, P. Aigle, B. Bode, H. B. Moreira, R. Li, Y. Luzhetskyy, A. Medema, M.
 H. Pernodet, J. L. Stadler, M. Tormo, J. R. Genilloud, O. Truman, A. W. Weissman, K.
 311 J. Takano, E. Sabatini, S. Stegmann, E. Brotz-Oesterhelt, H. Wohlleben, W., . . .

Muller, R. (2021). Towards the sustainable discovery and development of new
antibiotics. *Nat Rev Chem*, 5(10), 726-749. [https://doi.org/10.1038/s41570-021-](https://doi.org/10.1038/s41570-021-00313-1)
[00313-1](https://doi.org/10.1038/s41570-021-00313-1)

Mohammed-Naji, Q., Zulperi, D., Ahmad, K. & Hata, E. M. (2025). Nano formulation
development and antibacterial activity of cinnamon bark extract-chitosan
composites against Burkholderia Glumae the causative agent of Bacterial
Panicle Blight in rice. *Plos One*, 20(6), e0320032.
<https://doi.org/10.1371/journal.pone.0320032>

Ortega, L.& Rojas, C. M. (2021). Bacterial Panicle Blight and Burkholderia glumae:
From Pathogen Biology to Disease Control. *Phytopathology*, 111(5), 772-778.
<https://doi.org/10.1094/PHYTO-09-20-0401-RVW>

Ortega, L., Walker, K. A., Patrick, C., Wamishe, Y., Rojas, A. & Rojas, C. M. (2020).
Harnessing Pseudomonas protegens to Control Bacterial Panicle Blight of Rice.
*Phytopathology*, 110(10), 1657-1667. [https://doi.org/10.1094/PHYTO-02-20-](https://doi.org/10.1094/PHYTO-02-20-0045-R)
[0045-R](https://doi.org/10.1094/PHYTO-02-20-0045-R)

Peleg, A. Y., Seifert, H. & Paterson, D. L. (2008). Acinetobacter baumannii: emergence
of a successful pathogen. *Clin Microbiol Rev*, 21(3), 538-582.
<https://doi.org/10.1128/CMR.00058-07>

Prescott, J. F. (2017). History and Current Use of Antimicrobial Drugs in Veterinary
Medicine. *Microbiol Spectr*, 5(6). [https://doi.org/10.1128/microbiolspec.ARBA-](https://doi.org/10.1128/microbiolspec.ARBA-0002-2017)
[0002-2017](https://doi.org/10.1128/microbiolspec.ARBA-0002-2017)

Russo, A., Bassetti, M., Ceccarelli, G., Carannante, N., Losito, A. R., Bartoletti, M.,
Corcione, S., Granata, G., Santoro, A., Giacobbe, D. R., Peghin, M., Vena, A.,
Amadori, F., Segala, F. V., Giannella, M., Di Caprio, G., Menichetti, F., Del Bono,
336 V., Mussini, C., Petrosillo, N., De Rosa, F. G., Viale, P., Tumbarello, M., Tascini,
C., Viscoli, C. & Venditti, M. (2019). Bloodstream infections caused by
carbapenem-resistant Acinetobacter baumannii: Clinical features, therapy and
outcome from a multicenter study. *Journal of Infection*, 79(2), 130-138.
<https://doi.org/https://doi.org/10.1016/j.jinf.2019.05.017>

Sasaki, O., Igarashi, Y., Saito, N. & Furumai, T. (2002). Watasemycins A and B, new
antibiotics produced by Streptomyces sp. TP-A0597. *J Antibiot (Tokyo)*, 55(3),
249-255. <https://doi.org/10.7164/antibiotics.55.249>

Sati, H.Carrara, E.Savoldi, A.Hansen, P.Garlasco, J.Campagnaro, E.Boccia, S.Castillo-
Polo, J. A.Magrini, E.Garcia-Vello, P.Wool, E.Gigante, V.Duffy, E.Cassini,
346 A.Huttner, B.Pardo, P. R.Naghavi, M.Mirzayev, F.Zignol, M.Cameron,
347 A.Tacconelli, E.Aboderin, A.Al Ghoribi, M.Al-Salman, J.Amir, A., . . . Umubyeyi
Nyaruhirira, A. (2025). The WHO Bacterial Priority Pathogens List 2024: a
prioritisation study to guide research, development, and public health strategies
against antimicrobial resistance. *Lancet Infect Dis*. [https://doi.org/10.1016/S1473-](https://doi.org/10.1016/S1473-3099(25)00118-5)
[3099\(25\)00118-5](https://doi.org/10.1016/S1473-3099(25)00118-5)

Schar, D., Klein, E. Y., Laxminarayan, R., Gilbert, M. & Van Boeckel, T. P. (2020). Global
trends in antimicrobial use in aquaculture. *Sci Rep*, 10(1), 21878.
<https://doi.org/10.1038/s41598-020-78849-3>

Sundin, G. W.& Wang, N. (2018). Antibiotic Resistance in Plant-Pathogenic Bacteria.
*Annu Rev Phytopathol*, 56, 161-180. [https://doi.org/10.1146/annurev-phyto-](https://doi.org/10.1146/annurev-phyto-080417-045946)
[080417-045946](https://doi.org/10.1146/annurev-phyto-080417-045946)

Tavares, M., Kozak, M., Balola, A. & Sa-Correia, I. (2020). Burkholderia cepacia
Complex Bacteria: a Feared Contamination Risk in Water-Based Pharmaceutical
Products. *Clin Microbiol Rev*, 33(3). <https://doi.org/10.1128/CMR.00139-19>
Umemura, K., Kon, T., Suzuki, H., Kaneda, Y. & Iwasaka, T. (2021). Total synthesis of
antibiotic thiazostatin and watasemycin. *Studies in Science and Technology*,
10(1), 61-64. <https://doi.org/10.11425/sst.10.61>
Van Boeckel, T. P., Glennon, E. E., Chen, D., Gilbert, M., Robinson, T. P., Grenfell, B. T.,
Levin, S. A., Bonhoeffer, S. & Laxminarayan, R. (2017). Reducing antimicrobial
use in food animals. *Science*, 357(6358), 1350-1352.
<https://doi.org/10.1126/science.aao1495>
Velazquez-Meza, M. E., Galarde-Lopez, M., Carrillo-Quiroz, B. & Alpuche-Aranda, C. M.
(2022). Antimicrobial resistance: One Health approach. *Vet World*, 15(3), 743-
749. <https://doi.org/10.14202/vetworld.2022.743-749>
Verhaegen, M., Bergot, T., Liebana, E., Stancanelli, G., Streissl, F., Mingeot-Leclercq,
372 M. P., Mahillon, J. & Bragard, C. (2023). On the use of antibiotics to control plant
pathogenic bacteria: a genetic and genomic perspective. *Frontiers in*
*Microbiology*, 14, 1221478. <https://doi.org/10.3389/fmicb.2023.1221478>
WHO. (2023). *Antimicrobial resistance*. World Health Organization. Retrieved 7.21.2025
from <https://www.who.int/news-room/fact-sheets/detail/antimicrobial-resistance>

**Figure legends**

**Figure 1. The *P. protegens* PBL3 secretome inhibit the growth of a broad range of**
**plant pathogenic bacteria.** Bacterial strains OD₆₀₀=0.2 were grown on King's B (KB)
broth (black bars) or KB amended with the *P. protegens* PBL3 secretome (gray bars)
and incubated at 28 °C with constant agitation. Bacterial growth (OD₆₀₀) was measured
after 18h. Bars represent the average bacterial growth, and the line graph above the bar
represents the standard deviation of bacterial growth for three replications. Students' t-
test was used to evaluate statistical significance when comparing bacterial growth in
non-amended versus amended media. The asterisk above comparing bars indicates
statistically significant differences with P-value=0.05.

**Figure 2. The *P. protegens* PBL3 secretome inhibits the growth of a broad range of**
**human pathogens.** Pathogenic gram-positive bacteria were grown in agar plates for 20
390 hours. Bacteria were scraped from a plate and diluted in Phosphate-buffered saline (PBS)

buffer to an optical density 600 (OD₆₀₀) of 0.2 and further diluted in growing media to a
final concentration of 1.5×10^6 CFU/mL. Twenty microliters of the *P. protegens* PBL3
secretome or PBS were mixed with different volumes of growing media to a final volume
of 100 μ l and grown for 18h at 37°C with constant agitation. After 18 h, 0.1% 2,3,5-
Triphenyl-tetrazolium chloride was added to the cultures. Absorbance at 470 nm was read
in a plate reader. Bar represents the average absorbance of bacteria minus the
absorbance of media alone, and the error bar represents the standard deviation of six
replicates. Asterisk suggests a difference between the PBS and PBL3 secretome with
level of significance set at $p=0.05$.

**Supplementary Figure 1: Screening the antimicrobial property of *P. protegens***
**PBL3 secretome against gram-negative human pathogens.** Pathogenic gram-
negative bacteria were grown in agar plates for 20 hours. Bacteria was scraped from a
plate and diluted in Phosphate-buffered saline (PBS) buffer to an optical density 600
(OD₆₀₀) of 0.2 and further diluted in growing media to a final concentration of 1.5×10^6
CFU/mL. Different volumes of the *P. protegens* PBL3 secretome or PBS were mixed with
different volumes of growing media to a final volume of 100 μ l and grown for 18h at 37°C
with constant agitation. After 18 h, 0.1% 2,3,5-Triphenyl-tetrazolium chloride was added
to the cultures. Absorbance at 470 nm was read in a plate reader. Bar represents the
average absorbance of bacteria minus the absorbance of media alone, and the error bar
represents the standard deviation of six replicates. Asterisk suggests a difference
between the PBS and PBL3 secretome, with a level of significance set at $p=0.05$.

**Supplementary Figure 2: Screening the antimicrobial property of *P. protegens***
**PBL3 secretome against gram-positive human pathogens.** Pathogenic gram-positive

bacteria were grown in agar plates for 20 hours. Bacteria were scraped from a plate and
 diluted in Phosphate-buffered saline (PBS) buffer to an optical density 600 (OD₆₀₀) of 0.2
 and further diluted in growing media to a final concentration of 1.5 X 10⁶ CFU/mL. Different
 volumes of the *P. protegens* PBL3 secretome or PBS were mixed with different volumes
 of growing media to a final volume of 100 µl and grown for 18h at 37°C with constant
 agitation. After 18 h, 0.1% 2,3,5-Triphenyl-tetrazolium chloride was added to the cultures.
 Absorbance at 470 nm was read in a plate reader. Bar represents the average
 absorbance of bacteria minus the absorbance of media alone, and the error bar
 represents the standard deviation of six replicates. Asterisk suggests a difference
 between the PBS and PBL3 secretome with a level of significance set at p=0.05.

**Table 1. Bacterial strains used in this study**

Bacterial strains	Reference/Source
Acinetobacter baumannii	Martin Conda-Sheridan, University of Nebraska Medical Center
Burkholderia cenocepacia CU0318	(Asselin et. al. 2016)
Burkholderia cenocepacia CU3094	(Asselin et. al. 2016)
Burkholderia cenocepacia CU3368	(Asselin et. al. 2016)
Burkholderia cenocepacia CU3370	(Asselin et. al. 2016)
Burkholderia cenocepacia CU3371-1	(Asselin et. al. 2016)
Burkholderia cenocepacia CU3371-2	(Asselin et. al. 2016)
Burkholderia cenocepacia CU6878	(Asselin et. al. 2016))
Burkholderia gladioli CU3082	(Asselin et. al. 2016)
Burkholderia gladioli CU3083	(Asselin et. al. 2016)
Burkholderia gladioli CU3891	(Asselin et. al. 2016)
Burkholderia glumae UAPB10	Yeshi Wamishe, University of Arkansas, Fayetteville
Burkholderia glumae UAPB11	Yeshi Wamishe, University of Arkansas, Fayetteville
Burkholderia glumae UAPB13	Yeshi Wamishe, University of Arkansas, Fayetteville
Burkholderia sp. O64a	Steven Beer, Cornell University

Enterococcus faecium	Food Processing Center, University of Nebraska-Lincoln
Erwinia amylovora	Elena Garcia, University of Arkansas, Fayetteville
Escherichia coli K 12	ATCC 10798
Escherichia coli O157: H7	U.S. Department of Agriculture- Agriculture Research Service
Listeria innocua	ATCC 33090
Pseudomonas aeruginosa PAO1	Jim Alfano, University of Nebraska- Lincoln.
Pseudomonas aeruginosa PA103	Jim Alfano, University of Nebraska- Lincoln.
Pseudomonas protegens PBL3	Ortega et. al., 2020
Salmonella typhi	Food Processing Center, University of Nebraska- Lincoln
Staphylococcus aureus	ATCC 25923
Xanthomonas axonopodis	Craig Rothrock, University of Arkansas, Fayetteville
Yersinia enterocolitica	Jim Alfano, University of Nebraska- Lincoln.

**Acknowledgement:** We would like to thank Dr Martin Conda Sheridan (University of
Nebraska Medical Center) and Dr Byron Chaves (Department of Food Science and
technology, University of Nebraska-Lincoln) for sharing protocols and bacterial strains
and to Dinithi De Silva for her valuable assistance during the experiments. This work
was supported by the Nebraska Research Initiative.

Figure 1. The *P. protegens* PBL3 secretome inhibit the growth of a broad range of plant pathogenic bacteria. Bacterial strains OD₆₀₀=0.2 were grown on King's B (KB) broth (black bars) or KB amended with the *P. protegens* PBL3 secretome (gray bars) and incubated at 28 °C with constant agitation. Bacterial growth (OD₆₀₀) was measured after 18h. Bars represent the average bacterial growth, and the line graph above the bar represents the standard deviation of bacterial growth for three replications. Students' t-test was used to evaluate statistical significance when comparing bacterial growth in non-amended versus amended media. The asterisk above comparing bars indicates statistically significant differences with P-value=0.05.

Figure 2. The *P. protegens* PBL3 secretome inhibits the growth of a broad range of human pathogens. Pathogenic gram-positive bacteria were grown in agar plates for 20 hours. Bacteria were scraped from a plate and diluted in Phosphate-buffered saline (PBS) buffer to an optical density 600 (OD₆₀₀) of 0.2 and further diluted in growing media to a final concentration of 1.5 X 10⁶ CFU/mL. Twenty microliters of the *P. protegens* PBL3 secretome or PBS were mixed with different volumes of growing media to a final volume of 100 µl and grown for 18h at 37°C with constant agitation. After 18 h, 0.1% 2,3,5-Triphenyl-tetrazolium chloride was added to the cultures. Absorbance at 470 nm was read in a plate reader. Bar represents the average absorbance of bacteria minus the absorbance of media alone, and the error bar represents the standard deviation of six replicates. Asterisk suggests a difference between the PBS and PBL3 secretome with level of significance set at p=0.05.

Reviewer 1

1. Indicate what "those" refers to more explicitly, such as "eight of the predicted secondary metabolites"

Response: We have revised the introduction and addressed the comment in lines 76-83.

2. You mention selecting six gram-negative bacteria; is the other strain of *E. coli* not included in the count?

Response: We appreciate the comment. We clarified the statement in lines 176-179.

Reviewer 2

1. It is suggested to clarify the role of "secretome" in the title, for example: "The secretome of *Pseudomonas protegens* PBL3 exhibits broad-spectrum antimicrobial activity against plant and human pathogenic bacteria."

Response: Thank you. We modified the title per your suggestion.

2. The abstract does not mention specific inhibition rates or names of key pathogens. It is recommended to add 1-2 representative results (e.g., 95% inhibition against *A. baumannii*).

Response: Revised the abstract with minor addition. Line 21-25

3. The introduction section can strengthen the research background: the introduction of *P. protegens* and its secondary metabolites can be more systematic, especially its relationship with known antibiotics.

Response: We appreciate the suggestion. We have included additional information of the predicted secondary metabolites in the *P. protegens* PBL3 genome as well as additional information on their known antimicrobial activities in lines 76-86.

4. The discussion section can strengthen the speculation on mechanisms. Although the compounds have not been identified, it is still possible to speculate on their potential targets by combining the known mechanisms of known metabolites (such as pyoverdine, DAPG).

Response: Revised discussion. Line 261-273

5. The specific composition of "Minimal 9 media" is not stated in the text. It is recommended to supplement it to facilitate the repetition of the experiment.

Response: Thank you. We provided the specific composition of Minimal 9 media in lines 107-109.

6. It is not clear whether a "bacteria-free secretome control" has been set up to rule out the impact of the secretome itself on OD600.

Response: All the experiments were done with bacteria-free secretome. As explained in materials and methods (line 111), the secretome was filter sterilized.

7. The text uses "%v/v"; it is recommended to unify it as "% (v/v)" or clarify whether it refers to the final concentration.

Response: Revised as %(v/v) to be consistent throughout.

8. The article does not explain why TTC staining was chosen and its linear relationship with the number of viable bacteria. It is recommended to add a brief explanation of the principle.

Response: We provided clarification per suggestion in lines 131-137.

9. Only "student's t-test" is mentioned, without specifying whether it is two-tailed, paired, or unpaired, and no specific p-value is provided.

Response: Clarification was provided in line 141.

10. Lack of representative strain names in Figure 1. The specific Burkholderia strain names are not labeled in the bar chart. It is recommended to indicate them in the figure or figure caption.

Response: Revised in the figure with each strain detailed mentioned in the figure.

11. The label "Gram-positive bacteria" in Figure 2 is incorrect; the figure contains Gram-negative bacteria (such as *A. baumannii*). It is recommended to correct the figure caption or reclassify them.

Response: Thank you for catching that. Figure legend was modified to include both Gram-negative and Gram-positive bacteria as well as the lab strain *E. coli* K12.

12. The expression of inhibition rate data is inconsistent, for example, "~30%", ">30%", "~95%". It is recommended to unify it into the form of "mean {plus minus} standard deviation".

Response: Thank you for the suggestion. We decided to leave the text as a percentage to keep the smooth flow but added means and standard deviations in supplementary tables.

13. Supplementary Figure 1-2 lacks significance markers, although the text mentions "Asterisk suggests a difference", the asterisk (*) is not shown in the figure and needs to be supplemented.

Response: Both the supplementary figures have the asterisk above the bar. We have revised and made it bold to ensure proper visualization.

14. Insufficient explanation of ineffective strains. For example, there is no inhibition of *P. aeruginosa* and *S. typhi*, and the possible mechanisms (such as efflux pumps, biofilms, etc.) are not discussed.

Response: A discussion on the possible mechanism of resistance has been added in the discussion section. Line:274-297

15. Not compared with known antibiotics. The lack of comparison of activity with antibiotics commonly used in current clinical or agricultural settings makes it difficult to evaluate their practical application potential.

Response: Thank you for the comment. Additional information of antibiotics used to control plant pathogenic bacteria is provided in lines 311-314

16. Inconsistent use of italics for strain names. For example, *Burkholderia* is sometimes not italicized; it is recommended to unify this throughout the text.

Response: Confirmed all strains are italicized.

17. The format of the references needs to be unified, page numbers are missing in some references (e.g., Sati et al., 2025), and the DOI format is incomplete in others.

Response: edited and verified all the DOIs

Re: Spectrum02669-25R1 (The secretome of the environmental bacterium *Pseudomonas protegens* PBL3 has broad-spectrum antimicrobial activity against plant and human pathogenic bacteria)

Dear Dr. Clemencia M. Rojas:

Your manuscript has been accepted, and I am forwarding it to the ASM production staff for publication. Your paper will first be checked to make sure all elements meet the technical requirements. ASM staff will contact you if anything needs to be revised before copyediting and production can begin. Otherwise, you will be notified when your proofs are ready to be viewed.

Sincerely,
Lindsey Burbank
Editor
Microbiology Spectrum